# The Effect of Endurance Training on Serum BDNF Levels in the Chronic Post-Stroke Phase: Current Evidence and Qualitative Systematic Review

**DOI:** 10.3390/jcm11123556

**Published:** 2022-06-20

**Authors:** Sara Górna, Katarzyna Domaszewska

**Affiliations:** Department of Physiology and Biochemistry, Poznan University of Physical Education, 61-871 Poznań, Poland; domaszewska@awf.poznan.pl

**Keywords:** stroke, BDNF, older adults, neurorehabilitation, physical activity

## Abstract

Research in modern neurorehabilitation focusses on cognitive and motor recovery programmes tailored to each stroke patient, with particular emphasis on physiological parameters. The objectives of this review were to determine whether a single bout of endurance activity or long-term endurance activity regulates exercise-dependent serum brain-derived neurotrophic factor (BDNF) levels and to evaluate the methodological quality of the studies. To assess the effectiveness of endurance exercise among patients in the chronic post-stroke phase, a systematic review was performed, including searching EBSCOhost, PEDro, PubMed, and Scopus for articles published up to the end of October 2021. The PRISMA 2020 outline was used, and this review was registered on PROSPERO. Of the 180 papers identified, seven intervention studies (comprising 200 patients) met the inclusion criteria. The methodological quality of these studies was evaluated by using the Physiotherapy Evidence Database (PEDro) criteria. The effect of exercise was evaluated in four studies with a single bout of endurance activity, two studies with long-term endurance activity, and one study with a single bout of endurance activity as well as long-term endurance activity. The results of our systematic review provide evidence that endurance exercise might augment the peripheral BDNF concentration in post-stroke individuals.

## 1. Introduction

The global clinical effect of stroke is related to 132.1 million disability-adjusted life years [1]. The increase in the number of cerebrovascular incidents around the world has mostly been associated with the population ageing phenomenon [2]. Statistical reports suggest that 40% of the population in Poland will be over 60 years old by 2050 [3]. It is important to note that during the COVID-19 pandemic, ischaemic stroke patients with COVID-19 have shown worse functional outcomes than ischaemic stroke patients without COVID-19 [4]. Moreover, an international observational study across 40 countries revealed that infection with severe acute respiratory syndrome coronavirus-2 (SARS-CoV-2), the causative agent of COVID-19, occurred in 3.9% of acute stroke incidents [5]. Physical activity (PA) is one of the factors that can decrease the severe course of infection and increase survival among older people during the COVID-19 pandemic [6].

Post-stroke individuals commonly experience negative sequelae in the motor, sensory, emotion, and cognitive systems [7,8]. Factors such as gender, ageing, and the presence of neurological conditions may modulate these abilities in humans [9,10,11]. Other authors [12,13] have pointed out that improving motor and cognitive skills in post-stroke individuals is an essential factor for their recovery.

Growing evidence indicates a direct relationship between performing physical exercise and improved brain function, including in numerous cognitive processes [14,15]. However, there are no available data regarding the appropriate intensity and duration of endurance effort that is the most effective to facilitate post-stroke neuroplasticity. It is noteworthy that a properly structured PA programme can counteract the declines connected with advanced ageing or brain damage [16,17]. The positive influence of endurance exercise is associated with reducing proinflammatory status [18], neuronal apoptosis [19], and microglial reactivity [20]. From a physiological point of view, endurance effort may influence post-stroke neuroplasticity, including synaptic plasticity, long-term potentiation, angiogenesis, and neurogenesis [21,22,23]. 

Endurance exercise is one of the key factors causing the release of brain-derived neurotrophic factor (BDNF) [24]. This protein is only 27 kDa, so BDNF in the blood is able to cross the blood–brain barrier freely [25]. The *BDNF* gene is located on the short arm of chromosome 11 (11p14.1) [26]. Some studies have suggested [27,28,29] that the BDNF level in the blood may reflect its level in the brain, and changing the levels of this exerkine in the blood is reflected by changes in its level in the nervous system. However, the importance of the brain as a source of BDNF might be overestimated relative to peripheral sources in the blood circulation from the brain [30]. It is well established that BDNF plays a significant role in the development of the nervous system and is involved in the development of glial cells and serotonergic, hippocampal, and cortical neurons, among other cells [31]. Peripheral BDNF secretion is also affected by other factors, such as sex, age, body composition [32], circadian rhythm [33], and tobacco consumption [34]. 

Recently, a number of clinical studies have reported the advantageous impact of various types of endurance activity (EA) on enhancing BDNF levels in the peripheral blood in post-stroke patients. However, several studies have revealed that BDNF remains unchanged after sessions of different forms of high-intensity training in sedentary or in adults who train regularly [35,36,37]. Exerkines, especially BDNF, help regulate neuronal metabolism [38], ensure the correct functioning of neurons [39,40], modulate neurotransmission [41], and induce plasticity [32,38,41]. Higher BDNF levels have been implicated in better recognition [42], spatial [43], and hippocampal functioning [44]. It has been proposed that BDNF has a significant impact on lipid and glucose metabolism regulation [45]. Current published data from animal models and clinical studies present a wealth of evidence that BDNF influences appetite [46], blood glucose levels [47], and insulin sensitivity [48,49]. Moreover, patients experiencing post-stoke depression present with lower serum BDNF levels [50,51]. Furthermore, different factors, such as a history of depression, functional and/or cognitive decline, other comorbidities, and poor family support, may also have an impact on post-stroke depression [52].

The purpose of this systematic review was to answer the following questions: (1) Is there any relationship between participating in endurance effort and the circulating BDNF concentration in adult post-stroke individuals? (2) Are the recent clinical trials examining the influence of EA on BDNF expression, including their methodological approach, reliable?

## 2. Materials and Methods

### 2.1. Protocol and Registration

Searches were performed from 7 September to 28 October 2021. This systematic review was prepared by following the current recommendations of the Preferred Reporting Items for Systematic Reviews and Meta-Analyses (PRISMA) 2020 and the Cochrane Handbook of Systematic Reviews of Intervention, version 6.2 [53,54]. In addition, the protocol of this paper was registered in PROSPERO, the international database of systematic reviews (protocol registration no. CRD42021276961).

### 2.2. Eligibility Criteria

Articles that met the PICO (P—participant, I—interventions, C—comparisons, O—outcomes) eligibility criteria were subjected to further analyses [53,55]. The participants included post-stroke adults. The type of intervention was a single bout of EA or long-term EA. The control group included post-stroke patients who did not participate in EA or participants with different EA intensities and durations. Moreover, the outcome criterion consisted of papers that investigated the peripheral BDNF concentration. Only studies that appeared in peer-reviewed journals with an impact factor (IF) were selected for analysis. Non-human studies, reviews and meta-analyses, letters to the editor, case reports, editorial comments, articles not written in English, guidelines, conference reports, and protocols of clinical trials were excluded. The PRISMA 2020 flow chart depicting the trial eligibility process for this systematic review is shown in Figure 1.

### 2.3. Search Strategy

An exhaustive electronic search was performed in four databases with no date limits: EBSCOhost, PEDro, PubMed, and Scopus. Titles, abstracts, and keywords were searched by using the following Medical Subject Headings (MeSH) terms: ‘BDNF’, ‘stroke’, ‘endurance training’, ‘ physical activity’, and ‘aerobic exercise’, crossed with the Boolean operators ‘AND’ and ‘OR’. The identified articles were copied into the EndNote 20 bibliographic reference manager (Clarivate Analytics, Thomson Reuters, Philadelphia, PA, USA), and duplicate publications were removed. The analysed studies assessed the effects of a single bout of EA or long-term EA on the peripheral BDNF concentration of post-stroke individuals and the clinical outcomes in these individuals.

### 2.4. Data Extraction

Details such as the methods and results of the selected studies were extracted by two reviewers (SG and KD) independently and blindly. All disagreements during the analysis were resolved through consensus. The key information for each paper, including the authors and their nationalities, its year of publication, the type of effort (single bout of EA or long-term EA), patient characteristics (sample size, age, post-stroke time, and percentage of women), intervention exercise protocol (type, duration, frequency, and intensity), and clinical outcomes (BDNF concentration and other outcomes), was extracted. Moreover, information about the journal in which each article had been published was collected and analysed: its IF (for the year 2020), 5-year IF, total citations (TC), and average citations per year, according to the Web of Science Core Collection (Clarivate Analytics, Thomson Reuters).

### 2.5. Data Analysis: Quality Assessment

The methodological quality of all of the included papers assessing the peripheral BDNF levels of post-stroke individuals was appraised by using the Physiotherapy Evidence Database (PEDro). The validity and reliability of this tool in rating the methodological quality and risk of bias of clinical research studies has been confirmed in several studies [56,57]. The PEDro scale comprises eleven items that assess the quality of a clinical trial, as described in the Delphi list [58]. The PEDro scale incorporates two more items (criterion 8 and 10) than the Delphi list. The PEDro scale includes 11 items with scores that range from 0 to 10 for items 2–11. On this scale, a paper’s quality can be classified as excellent (9–10), good (6–8), fair (4–5), or poor (<4). Subsequently, the Sackett grade system was used to describe the level of evidence; this grading system is recommended by evidence-based medicine guidelines [59]. According to this system, studies that obtained ≥6 more points on the PEDro scale were considered level 1 evidence, and studies that obtained <6 points on the PEDro scale were considered level 2 evidence.

Microsoft Excel (Redmond, WA, USA) and Statistica version 13.3 (TIBCO Software, Round Rock, TX, USA) were used to generate descriptive statistics, including the mean and standard deviation (SD). The following data were extracted from the studies: sample size, age, PEDro score, total citation, average citations, and journal impact factor (Table 1, Table 2 and Table 3). The most significant parameters of endurance effort in a single bout as well as long-term EA trials among post-stroke participants were considered. Changes in the peripheral BDNF levels and other clinical outcomes (lactate concentration, CSP response, IGF-1 concentration, ACER points, VO_2peak_, CO, HHb, THb, cell-bearing neurities, RPMP) obtained in assessments before and after a single bout of EA or long-term EA were specified to calculate the delta percent of variation (Δ%) (Table 4). The variables had a non-Gaussian distribution, so the Mann–Whitney U test was used (Table 4). The statistical significance level was set at *p* < 0.05 for all comparisons.

## 3. Results

### 3.1. Study Selection

In the first phase of database searches, 180 records related to the above-mentioned keywords and MeSH terms were found in PubMed (*n* = 120), PEDro (*n* = 3), Scopus (*n* = 32), and EBSCOhost (*n* = 25). In this stage, 46 duplicates were removed. Therefore, 134 articles were screened by reading the titles and abstracts. Out of these, 111 records were excluded from further analysis. The full texts of each of the 23 remaining articles were read. In the end, seven clinical trials fulfilled all of the inclusion criteria and were accepted for qualitative analysis in our systematic review. The current version of the modified PRISMA 2020 flow chart illustrates the study selection phases (Figure 1). All of the included papers were published after 2014, with six being published after 2018. The trials had been conducted in the United States (*n* = 2), Canada (*n* = 2), Brazil (*n* = 1), Egypt (*n* = 1), and Taiwan (*n* = 1).

### 3.2. Study Characteristics

A total of seven [60,61,62,63,64,65,66] clinical trials involving 200 patients (95 subjected to a single bout of EA, 53 subjected to long-term EA, and 52 subjected to a single bout of EA as well as long-term EA) were included in the systematic review. The characteristics of the papers are displayed in Table 1. The mean sample size was 28.57 ± 13.9 (23.75 ± 12.7 participants for studies including a single bout of EA and 26.5 ± 4.95 participants for studies including a long-term EA). Only one trial included more than 50 patients [60]. Regarding the EA duration, four studies (57.1%) investigated a single bout of activity, two studies (28.6%) investigated a regular programme of activity, and only one study investigated a single bout as well as long-term activity (14.3%). Females represented 33.5% of the patients (40.1% female participants for studies including a single bout of EA, 21.5% female participants for studies including long-term EA, and 30.8% female participants for studies including a single bout of EA as well as studies including long-term EA). The mean time since stroke was 54.6 ± 29.7 months. Four studies were randomised controlled trials (RCT), two were controlled clinical trials (CCT), and one was a prospective clinical trial (PCT). The mean age was 58.06 ± 6.51 years (60.27 ± 3.55 years for studies including a single bout of EA, 52.10 ± 5.23 years for studies including long-term EA, and 63.4 ± 11.3 years for studies including a single bout of EA as well as studies including long-term EA). Out of the seven studies, four [60,61,62,63] described power calculations to estimate the requisite sample size.

**Table 1 jcm-11-03556-t001:** Characteristics of the included studies.

No.	Type	Study	Country	Design	Patients (*n*)	Female (%)	Age (Years)	Post-Stroke Period (Months)
1.	Single bout	Charalambos et al., 2018 [65]	The United States	RCT	34	32.35%	60.47 ± 12.42	53.82 ± 57.42
2.		Morais et al., 2018 [63]	Brazil	CCT	10	50%	58.0 ± 12.8	110.4 ± 69.6
3.		Boyne et al., 2019 [61]	The United States	RCT	16	43.8%	57.4 ± 9.7	78 ± 49.2
4.		King et al., 2019 [66]	Canada	CCT	35	34.29%	65.2 ± 9.4	31.5 ± 26.7
5.	Long term	El-Tamawy et al., 2014 [64]	Egypt	PCT	30	30%	48.4 ± 6.39	3–18
6.		Hsu et al., 2020 [62]	Taiwan	RCT	23	13.04%	HIIT 58.5 (49.8–67.2) MICT 53.1 (46.2–60.0)	HIIT 38.5 (19.1–57.9) MICT 28.8 (3.35–54.2)
7.	Single bout and long term	Ploughman et al., 2019 [60]	Canada	RCT	52	3.77%	63.4 ± 11.3	41.0 ± 39.8

BDNF—brain-derived neurotrophic factor; RCT—randomized controlled trial; CCT—clinical controlled trial; PCT—prospective clinical trial; HIIT—high-intensity interval training; MICT—moderate-intensity continuous training.

### 3.3. PEDro Assessment and Study Quality

The mean 2020 journal IF was 3.03 ± 1.11 (2.55 ± 0.67 for studies including a single bout of EA, 3.59 ± 1.97 for those including long-term EA, and 3.92 for studies including a single bout of EA as well as studies including long-term EA). The mean 5-year IF was 3.65 ± 1.36 (3.02 ± 0.67 for studies including a single bout of EA, 4.06 ± 2.21 for studies including long-term EA, and 5.38 for studies including a single bout of EA as well as studies including long-term EA). The mean TC was 17.14 ± 19.98 (10.75 ± 3.78 for studies including a single bout of EA, 31.0 ± 42.43 for studies including long-term EA, and 15 for studies including a single bout of EA as well as long-term EA). The mean TC for both EA duration types was not significantly different. However, the mean TC was slightly higher for studies including long-term EA. The mean citations per year for all of the assessed trials was 3.73 ± 2.24 (3.13 ± 1.21 for studies including a single bout of EA, 4.32 ± 4.69 for studies including long-term EA, and 5 for studies including a single bout of EA as well as long-term EA) (Table 2).

**Table 2 jcm-11-03556-t002:** Clinical trials on endurance exercise and post-stroke BDNF concentration and their quality evaluation based on the Web of Science Core Collection (Clarivate Analytics).

No.	Type	Study	Journal	2020 Journal Impact Factor	5 year Journal Impact Factor	TC	AC
1.	Single bout	Charalambos et al., 2018 [65]	Top Stroke Rehabil.	2.119	2.797	8	2.0
2.		Morais et al., 2018 [63]	Top Stroke Rehabil.	2.119	2.797	14	3.5
3.		Boyne et al., 2019 [60]	J Appl Physiol.	3.531	4.006	14	4.67
4.		King et al., 2019 [66]	Neurol Res.	2.448	2.480	7	2.33
5.	Long term	El-Tamawy et al., 2014 [64]	Neuro-Rehabilitation	2.138	2.501	61	7.63
6.		Hsu et al., 2020 [62]	Ann Phys Rehabil Med.	4.919	5.622	1	1.0
7.	Single bout and long term	Ploughman et al., 2019 [60]	Neurorehabil Neural Repair.	3.919	5.378	15	5.0

TC—total citations; AC—average citations; n/a—not available.

The average quality assessment using the PEDro criteria of the included studies was 4.29 ± 2.21. For studies including a single bout of EA, the mean PEDro score was 3.25 ± 1.5 (range 2–5), and for studies including long-term EA, it was 4.5 ± 2.12 (range 3–6). Only two studies [60,62] were classified as level 1 evidence (Table 3).

**Table 3 jcm-11-03556-t003:** Quality assessment of the included studies using the PEDro evaluation criteria.

No.	Type	Study	PEDro Criteria												EL
			C1	C2	C3	C4	C5	C6	C7	C8	C9	C10	C11	TS *	
1.	Single bout	Charalambos et al., 2018 [65]	+	+	-	-	-	-	-	+	-	+	+	4	2
2.		de Morais et al., 2018 [63]	+	-	-	-	-	-	-	+	-	-	+	2	2
3.		Boyne et al., 2019 [61]	+	+	-	-	-	-	+	+	-	+	+	5	2
4.		King et al., 2019 [66]	-	-	-	-	-	-	-	+	-	-	+	2	2
5.	Long-term	El-Tamawy et al., 2014 [64]	+	-	-	+	-	-	-	-	-	+	+	3	2
6.		Hsu et al., 2020 [62]	+	+	-	+	+	-	+	-	-	+	+	6	1
7.	Single bout and long-term	Ploughman et al., 2019 [60]	+	+	+	+	-	-	+	+	+	+	+	8	1

Legend for PEDro criteria: C1, eligibility criteria were specified; C2, subjects were randomly allocated to groups; C3, allocation was concealed; C4, the groups were similar at baseline regarding the most important prognostic indicators; C5, there was blinding of all subjects; C6, there was blinding of all therapists who administered the therapy; C7, there was blinding of all assessors who measured at least one key outcome; C8, measures of at least one key outcome were obtained from more than 85% of the subjects initially allocated to groups; C9, all subjects for whom outcome measures were available received the treatment or control condition as allocated or, where this was not the case, data for at least one key outcome were analysed by ‘intention to treat’; C10, the results of between-group statistical comparisons are reported for at least one key outcome; C11, the study provides both point measures and measures of variability for at least one key outcome. Abbreviations: PEDro—Physiotherapy Evidence Database; TS—total score; EL—evidence level; *—criteria 2–11 scored.

### 3.4. Effect of a Single Bout of Endurance Activity on Serum BDNF Concentrations

Five trials [60,61,63,65,66] assessed the change in the serum BDNF concentration immediately after a single bout of EA in post-stroke individuals (Table 4). Among these studies, the active phase of EA ranged from 5 to 30 min. The most common type of activity was treadmill activity with or without body-weight support (BWS). The intensity of effort varied among the analysed studies. Only studies that involved ≥20 min of moderate- or high-intensity EA showed significant changes [61,63]. Three studies characterised the cognitive status of participants [60,63,66].

### 3.5. Effect of Long-Term Endurance Activity on Serum BDNF Concentrations

Three trials [60,62,64] described the change in the serum BDNF concentrations in post-stroke individuals participating in regular EA, with a duration ranging from 8 to 12 weeks (Table 4). The active phase of EA ranged from 20 to 30 min, and the frequency was 2–3 sessions per week. Two studies [62,64] evaluated the EA effort using a bicycle ergometer, and one study [60] used a treadmill with BWS. Only one [60] study investigated the combined effect of aerobic exercise with cognitive training. Moreover, two studies [60,64] used aerobic exercise for the study group, and one study [62] used a high-intensity interval training (HIIT) protocol. The studies that evaluated the effect of endurance training on peripheral BDNF levels showed variable cognitive abilities among the participants. One study [64] included people with cognitive impairment, another [62] showed participants without cognitive impairment, and in the third study [60], 45% of the included post-stroke individuals had scores below normal cognition. Only one study (by Ploughman) conducted a 12-week follow-up analysis of BDNF concentrations. Two studies [62,64] indicated a significant change in serum BDNF concentrations after finishing the EA programme.

**Table 4 jcm-11-03556-t004:** Methodological characteristics and main results of studies assessing the effects of endurance exercise on the BDNF concentration post-stroke.

No.	Type	Study	Protocol					Results	
			Outcome Measures	Groups	Active Phase of Exercise	Dose, Intensity	Type of Intensity	Serum BDNF (ng/mL)	Clinical MeasuresScale
1.	Single bout	Charalambos et al., 2018 [65]	BDNF Lactate	(1) CON (2) TMW (3) TBE	5 min	(1) Walking at 25% of their fastest comfortable speed (2) High-intensity range (70–85% HR_max_) or 13–15 RPE speed increased gradually, 0.05m/sek every 15 s (3) High-intensity range (70–85% HR_max_) or 13–15 RPE speed increased gradually, 0.05m/sek every 15 s	(1) Treadmill (2) Treadmill (3) Cycled on a total body exerciser	(1) NS (2) NS (3) NS No significance within or between groups	Lactate (mM/L) (1) ↑Δ = 0.16 (2) ↑Δ = 2.22 (3) ↑Δ = 6.10 A significant increase between post and pre levels
2.	Single bout	Morais et al., 2018 [63]	BDNF	(1) Mild intensity (2) Moderate intensity	30 min	(1) 50–63% HR_max_ (2) 64–76% HR_ma_	(1) Walk (2) Walk	(1) Δ = −0.04 (2) ↑Δ = +0.05 Significant differences between post and pre at moderate intensity. No significance between post vs. pre levels at mild intensity.	-
3.	Single bout	Boyne et al., 2019 [61]	BDNF CSP	(1) GXT (2) HIT—treadmill (3) HIT—stepper (4) MCT—treadmill	(1) Symptom-limited (2) 20 min (3) 20 min (4) 20 min	(1) Incline increased 2–4% every 2 min; (2) and (3) 30 s burst of max speed walking alternated with 30-to-60-s recovery periods; (4) 45 ± 5% HRR.	(1) Treadmill (2) Treadmill (3) Stepper (4) Treadmill	(1) Δ = +4.6 (2) Δ = +3.2 (3) Δ = +2.1 (4) Δ = −0.7 Statistically differences between post levels at 2 vs. 4 Δ= +3.9	CSP response from T0 to T20 (1) NS (2) ↑Δ = −0.1% (3) ↑Δ = +0.2% (4) ↑Δ = +2.9%
4.	Single bout	King et al., 2019 [66]	BDNF IGF-1	GXT	Symptom-limited	0% treadmill grade for the initial 2 min, followed by a 2.5% increase in grade every 2 min until an incline of 10% was reached and a 0.05 m/s increase in speed every 2 min thereafter until the test was terminated	Body-weight-supported treadmill or total-body recumbent stepper	Δ = +2.0	IGF-1 ↓Δ = −0.98 ng/mL
5.	Long term	El-Tamawy Long termet al., 2014 [64]	BDNF ACER	(1) Control group (2) Study group	8 weeks (1) 25–30 min 3 days/wk (2) 30 min 3 days/wk;	(1) Physiotherapy program (stretching and strengthening exercises, facilitation for each muscle, postural control, balance, functional and gait training) (2) and physiotherapy program + aerobic exercise; intensity not stated	(1) - (2) Bicycle ergometer	(2) post vs. pre ↑Δ = +4.65 post (2) vs. (1) ↑Δ = +3.17	ACER post (2) vs. (1) ↑Δ = 5.14
6.	Long term	Hsu et al., 2020 [62]	BDNF VO_2peak_ CO AVO_2diff_ O_2_Hb HHb THb MMSE cell-bearing neurities	(1) HIIT (2) MICT	36 sessions 2–3 sessions week	(1) Five 3 min intervals at 80% VO_2peak_ with each interval separated by 3 min of exercise at 40% of VO_2peak_ (2) 60% VO_2peak_	Bicycle ergometer	(1) ↑Δ= + 1.85 (2) ↓Δ = −1.42	VO_2peak_ (1) ↑Δ = +17.2% (2) ↑Δ = +8.09% CO (1) ↑Δ = +14.8% (2) NS AVO_2diff_ NS O_2_Hb NS HHb (1) ↑Δ = +55.8% (2) NS THb (1) ↑Δ = +47.0% (2) NS MMSE NS cell-bearing neurities (1) ↑Δ = +14.2% (2) NS
7.	Single bout and long term	Ploughman et al., 2019 [60]	BDNF IGF-1 RPMT	(1) Aerobic + COG (2) Aerobic + Games (3) Activity + COG (4) Activity + Games	10 weeks 20–30 min aerobic or activity; 20–30 min COG or games	(1) and (2) 60–80% of VO_2peak_ (3) and (4) therapeutic activity, functional task training, intensity not stated	(1) and (2) Treadmill with body weight support (3) - (4) -	pre vs. post NS pre vs. follow-up NS	IGF-1 NS across groups RPMP pre vs. follow-up (1) ↑Δ = +5.7 vs. (4) ↓Δ = −2.1

BDNF—brain-derived neurotrophic factor; CON—control group; CSP—cortical silent period; GXT—graded exercise test; HIT—high-intensity training; MCT—moderate continuous training; HRR—heart rate recovery; HR_max_—maximum heart rate; T0—before the warm-up; T20—after exercise; RPE—Rate of Perceived Exertion Scale; ACER—Addenbrooke’s Cognitive Examination—Revised; VO_2max_—peak oxygen consumption; CO—cardiac output; AVO_2diff_—arteriovenous O_2_ difference; O_2_Hb—oxyhaemoglobin; HHb—deoxyhaemoglobin; THb—total haemoglobin; MMSE—mini-mental status examination; TMW—treadmill walking; TBE—total-body exercise; COG—computerized dual working memory training; RPMT—Raven’s Progressive Matrices Test; ↑—statistically significant increase; ↓—statistically significant decrease.

## 4. Discussion

Exercise-induced changes in peripheral BDNF expression after a single bout of EA or chronic EA have been related to improved brain function [67]. It is also worth emphasising that the interplay between cognition and motor function is essential in the active recovery process after stroke [12,13,68]. A factor that may impact BDNF expression is the difference in cognitive status among the individuals included in the trials. Indeed, only one of the trials [64] included in this review solely examined participants with cognitive impairment in the chronic post-stroke period. The authors reported a positive correlation between the increased serum BDNF levels and cognitive function, namely in attention and visual spatial capacity, during a systematic 8-week aerobic training programme [64]. In another study [60], less than half of the participants were classified as being cognitively normal. The authors indicated that 10 weeks of aerobic exercises with cognitive training improved fluid intelligence but did not change peripheral BDNF levels [60]. Hsu et al. [62] examined 23 stroke patients without cognitive impairment and showed that longitudinal HIIT enhanced BDNF expression significantly more than moderate-intensity continuous training (MICT). Only one trial [60] in this systematic review indicated that endurance exercise without additional cognitive training did not enhance fluid intelligence in post-stroke patients. Tang et al. [69] and Bo et al. [70] confirmed this finding. However, Moore et al. [71] and Quaney et al. [72] demonstrated that long-term endurance training improved executive function and motor learning during the chronic phase of post-stroke rehabilitation in an RCT. The recovery of motor function after stroke requires motor abilities to be relearned through neuroplasticity [73]. Furthermore, the exercise-induced elevation of the serum BDNF was related to increased brain oxygenation [74]. In addition, the clinical assessment scales used to examine cognitive functions had an impact on the differences observed after EA sessions.

Only two of the included studies [60,62] were classified as level 1 evidence. These papers evaluated either the combined effects of a single bout of EA as well as long-term EA [60] or the long-term effect of EA [60]. None of the trials received a score for criteria 5 (blinding of all subjects) or 6 (blinding of all therapists who administered the therapy). Six of seven papers had been published within the last 5 years. An article published in 2014 [64] had been cited the most out of the included studies. This is probably because it was a pioneering article in the field of endurance exercise, cognitive function, and peripheral BDNF expression in post-stroke individuals and because it was published four years before the second-oldest study. Research in this area is still in the scientific inquiry phase, with efforts being directed towards identifying the best method to support neurorehabilitation.

In a systematic review of 20 studies on animal models of stroke, Alcantara et al. [75] reported that aerobic activity supports changes in the central BDNF concentration. A recent study involving animal models ascertained that 30 min of daily low-intensity systematic exercise improves cognitive abilities (for example, spatial learning, location task, as well as object recognition). Such positive changes were associated with higher BDNF production in the hippocampus [76,77]. Moreover, the higher levels of stress hormones induced during HIIT may downregulate hippocampal BDNF levels in rats [78]. However, the data obtained from animal models should be supported by clinical trials [79].

Recent systematic reviews with meta-analyses have revealed that peripheral BDNF is upregulated following a single session of moderate- and high-intensity exercise in healthy subjects [67,71]. This result confirms the findings reported by de Morais et al. [63] and Boyne et al. [61] in post-stroke individuals. Contrasting studies performed by Charlambous et al. [65], King et al. [66], and Ploughman et al. [60] reported that serum BDNF levels remain constant after a single bout of endurance exercise or a graded exercise test (GXT). An increase in the serum BDNF expression during endurance exercise with sufficient intensity could support elevated corticospinal excitability and central motor activation [61]. It is plausible that the key factors promoting serum BDNF secretion following a single bout of exercise are the intensity of the exercise (moderate or high) and its duration (>20 min).

We noted no activity-dependent upregulation in the peripheral BDNF in two studies [61,66] after GXT and in one study with 5 min of activity of different intensities [65]. The mean GXT duration was not sufficient to change the BDNF expression, even though this duration had been recommended in other exercise protocols [11,63]. Another study among 50 adults performing acute cycling exercises (duration of 45–55 min) indicated that an exercise-induced increase in BDNF concentration could improve memory [80]. A recent meta-analysis of 14 studies in healthy adults confirmed a significant peripheral increase in BDNF after a single bout of exercise [11]. A single bout of aerobic exercise might facilitate improved neurological recovery [73]. Current scientific evidence indicates that a single high-intensity bout of exercise could enhance cognition and motor skills, as they pertain to performing upper-extremity tasks, and promote post-stroke recovery in humans [73,81]. Such effects may be related to elevated levels of peripheral neurophysiological biomarkers such as BDNF and lactate [81,82]. Moreover, Thomas et al. [83] confirmed that a single HITT session combined with motor task learning has a greater effect on long-term motor skills than MICT. On the contrary, clinical trials have indicated that a single bout of MICT is more effective than HIIT at improving performance in a learning memory task [84]. Moreover, Tang et al. [69] and Hasan et al. [85] suggest that there is inconclusive evidence as to whether aerobic exercise could improve cognition in post-stroke individuals. Indeed, a single bout of aerobic exercise promoted a ‘mostly transient’ exercise-evoked change in peripheral BDNF levels in most people with chronic health dysfunctions [86]. Furthermore, it is necessary to study how different endurance exercise modalities affect brain plasticity [87].

In a systematic review representing data from nine studies and a total of 215 participants following a regular 8-week EA training protocol, Limaye et al. [88] showed that this type of activity increases BDNF levels. Endurance exercise may facilitate post-stroke recovery by enhancing cardiovascular fitness, thereby helping the individual to participate in PA [89,90]. This process may be due to the upregulation of the mBDNF/proBDNF ratio and altered neural plasticity [91]. In a review of 14 papers with 736 post-stroke participants, Oberlin et al., 2019 [92] suggested that combined aerobic and strength training yielded the highest cognitive performance gains. Endurance training also improved balance, motor control, and gains in hippocampal volume [83,93].

A recent meta-analysis with meta-regression reinforces the evidence that the BDNF expression is correlated with PA intensity [94]. In several of the papers included in this systematic review, the researchers described the endurance exercise intensity as being at the maximum oxygen consumption rate (% VO_2max_) [60,62] or maximum heart rate (HR_max_) measured from expired air with a facemask interface or estimated by an age-predicted formula [61,63,65]. Moreover, one study failed to describe exercise intensity [62,64]. In our systematic review, as well as in another review written by Knaepen et al. [86], the intensity threshold for BDNF upregulation was >70% HR_max_. In our review, only one RCT had an intensity above 60% HR_max_ and did not present a significant response in terms of the BDNF expression [60]. Additionally, two studies specified the exercise intensity using the blood lactate concentration [61,65].

Patients in the chronic phase of post-stroke rehabilitation may be able to safely add HIIT to their exercise program. Compared to moderate-intensity training, HIIT leads to a higher HR in post-stroke individuals [95]. Endurance exercise regimens with HIIT could positively impact brain and cardiovascular health and could be a part of cardiopulmonary rehabilitation in post-stroke patients [61,96,97]. Moreover, this type of training is feasible and recommended in post-stroke patients who do not present with any symptoms of exercise intolerance [96,98]. Other researchers claim that exercise with gradually increasing treadmill intensity might be able to improve their motor control, enhance their hippocampal BDNF secretion, and lower their stress more effectively compared to exercise practised at a constant intensity [99].

This change in the exercise-induced BDNF levels is consistent with blood lactate accumulation [100]. Indeed, changes in peripheral lactate accumulation have been associated with the upregulation of peripheral and central BDNF expression, motor cortex excitability, and brain metabolism [101,102,103]. Among stroke survivors, a mean blood lactate concentration of >4.7 mmol/L after 20 min of exercise was sufficient to promote a significant increase in BDNF concentration [61]. Moreover, when participants with a mean blood lactate concentration of 3.0 mmol/L performed 5 min of HIIT, an increase in BDNF levels was also observed [61]. On the contrary, Charalambous et al. [65] showed that a mean blood lactate concentration of 6.1 mmol/L after 5 min of exercise did not change the serum BDNF concentration in post-stroke patients. Significantly greater changes in the corticospinal excitability and peripheral BDNF were elicited after a single high-intensity session but not after a single MCT session [61]. Therefore, trials are needed to evaluate the physiologic influence of exercise intensity parameters such as lactate threshold or VO_2 peak_ on the cognitive status of individuals with neurological disorders in different phases of disability [12].

There is an urgent need to focus on comorbid conditions in post-stroke patients and their relationship with the BDNF levels. Further, follow-up assessments could help researchers and clinicians to understand whether there are any long-lived and continued benefits of a single bout of EA or long-term EA performed during the chronic phase of post-stroke rehabilitation. Subsequent trials should include more detailed assessments to determine whether BDNF changes coincide with other biomarkers, as well as cognitive and functional tests.

Of note, only one systematic review [75] has directly assessed the effects of PA and post-stroke BDNF concentration. However, the paper mostly focussed on animal models, such as those using Sprague Dawley rats, Wistar rats, and Mongolian gerbils. There is a gap in the clinical evidence regarding the most appropriate endurance exercise algorithm to facilitate neuroplasticity in post-stroke patients. It seems necessary to design a standard procedure for planning personalised endurance training that promotes neuroplasticity in post-stroke individuals. Future research concerning the effects of endurance exercise on BDNF in post-stroke individuals might focus on other medical conditions (e.g., diabetes, depression, and obesity) and evaluate BDNF changes across disease conditions.

## 5. Conclusions

In our up-to-date systematic review of the effects of EA on BDNF expression, we found that the majority of trials have a limited methodological design with low levels of evidence. Our review fills the gap concerning studies evaluating the effects of a single bout of EA and long-term EA on exercise-induced BDNF release to shed light on neurorehabilitation research. In conclusion, the present data indicate that endurance exercise might stimulate peripheral BDNF secretion in the chronic post-stroke phase. However, such results require further high-quality, randomised, prospective trials with follow-up research to establish the best EA protocol. This endeavour would help to confirm improvements in the neuroplastic mechanisms that might improve cognitive and functional performance in post-stroke individuals.

## Figures and Tables

**Figure 1 jcm-11-03556-f001:**
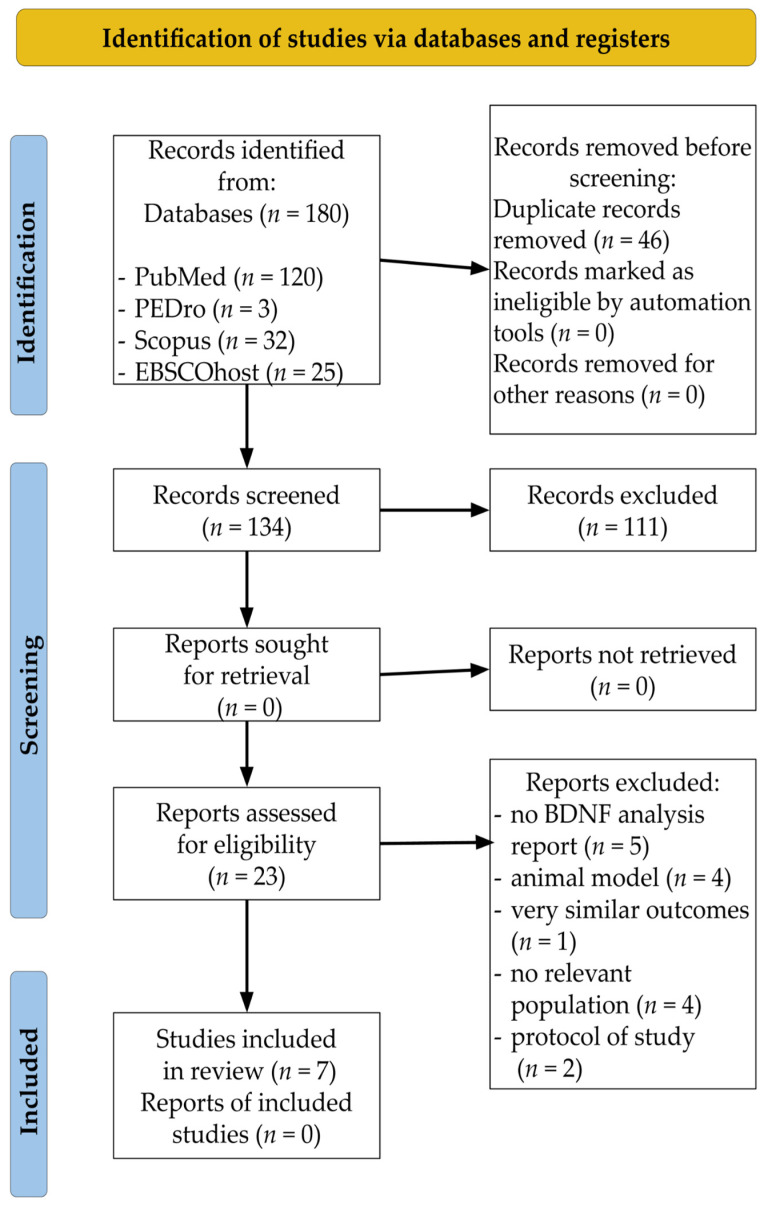
PRISMA 2020 flow chart of the trial selection process.

## Data Availability

Not applicable.

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
