# Peer review of "The Effect of Endurance Training on Serum BDNF Levels in the Chronic Post-Stroke Phase: Current Evidence and Qualitative Systematic Review"

_jcm, 2022, doi:10.3390/jcm11123556_

Round 1

Reviewer 1 Report

The paper describes that the effect of endurance training on serum BDNF levels in the chronic post-stroke phase. The authors also investigated whether a single bout of endurance activity or long-term endurance activity regulates exercise-dependent serum brain-derived neurotrophic factor (BDNF) concentrations and evaluated the methodological quality of the studies.

The paper is interesting and the results are novel. However, I have a few comments that the authors should consider, as described below:

1. Why does the authors focus on BDNF in serum rather than plasma? Platelets contain part of BDNF, which is not in serum. Although BDNF is highly concentrated in the nervous system, it is also found in the serum of humans and other mammals. It has been shown that this blood BDNF is essentially stored in blood platelets, from which it can be released into the plasma through activation or clotting processes[1].Low brain-derived neurotrophic factor (BDNF) levels in serum of depressed patients probably results from lowered platelet BDNF release.

[1]    Fujimura H, Altar CA, Chen R, et al. Brain-derived neurotrophic factor is stored in human platelets and released by agonist stimulation. Thromb Haemost. 2002;87(4):728-734.

2. The Introduction of the paper can be simplified.

For example, “The activation and proper functioning of the BDNF signalling pathway in the brain and other organs requires proteins such as cathepsin B, irisin, insulin-like growth factor 1 (IGF-1), peroxisome proliferator-activated receptor gamma coactivator 1-alpha (PGC1α), beta-hydroxybutyrate (BHB), and vascular endothelial growth factor (VEGF)”. It is not related to the main study aim of the article and can be deleted.

3. The introduction needs to be restructured. For example, the function of BDNF is described in the sixth paragraph, but it is mentioned in the preceding paragraphs.

4. In the third paragraph of introduction, the authors state that This protein is only 27 kDa, so BDNF in the blood is able to cross the blood-brain barrier freely”. In human neurons, the BDNF transcript is translated into pre-pro-BDNF in the neuronal cell body, which is cleaved into the precursor pro-BDNF. Pro-BDNF is both converted intracellularly and secreted extracellularly. The authors should indicate whether serum BDNF includes precursor BDNF.

5. The authors should elaborate more on the relationship between BDNF in serum and BDNF in brain.

6. The authors state that “There is a positive correlation between the BDNF concentration in the brain and the serum. A BDNF study in a rat model revealed that the correlation between its concentrations in the cortex and serum was > 0.8.Please provide references.

7. The authors state that “Recently, a number of clinical studies have reported the advantageous impact of various types of endurance activity (EA) on enhancing BDNF levels in peripheral blood in post-stroke patients.” Please provide references.

Minor:

Figure1: The picture is not clear enough.

Inconsistent annotation font: TABLE2: “TC - total citation by Web of Science Core Collection, AC - average citations per year by Web of Science Core. Collection, n/a - not available.

The average quality assessment using the PEDro criteria of the included studies was 4.29 ± 2.21.

Author Response

Thank you for giving us the opportunity to submit a revised draft of the manuscript ‘The Effect of Endurance Training on Serum BDNF Levels in the Chronic Post-Stroke Phase: Current Evidence and Qualitative Systematic Review’ for publication in the Journal of Clinical Medicine. We appreciate the time and effort that you dedicated to providing feedback on our manuscript and are grateful for the insightful comments on and valuable improvements to our paper. We have incorporated most of the suggestions made by you. Those changes are highlighted within the manuscript. Please see below and in the manuscript, in yellow, for a point-by-point response to your comments and concerns.

  1. Why does the authors focus on BDNF in serum rather than plasma? Platelets contain part of BDNF, which is not in serum. Although BDNF is highly concentrated in the nervous system, it is also found in the serum of humans and other mammals. It has been shown that this blood BDNF is essentially stored in blood platelets, from which it can be released into the plasma through activation or clotting processes[1].Low brain-derived neurotrophic factor (BDNF) levels in serum of depressed patients probably results from lowered platelet BDNF release.

[1]    Fujimura H, Altar CA, Chen R, et al. Brain-derived neurotrophic factor is stored in human platelets and released by agonist stimulation. Thromb Haemost. 2002;87(4):728-734.

 The authors focus on BDNF in serum because plasma BDNF represents freely BDNF and thus may have a different physiological role from serum BDNF. [28] Lommatzsch 2005

Plasma BDNF levels decrease with age and body mass gain, as opposed to the constant concentration of BDNF in serum and platelets. [28 and Begliuomini et al. 2007]

Begliuomini, S.; Casarosa, E.; Pluchino, N.; Lenzi, E.; Centofanti, M.; Freschi, L.; et al. Influence of endogenous and exogenous sex hormones on plasma brain-derived neurotrophic factor. Hum. Reprod. 2007, 22, 995–1002. doi: 10.1093/humrep/del479.

  1. The Introduction of the paper can be simplified.

For example, “The activation and proper functioning of the BDNF signalling pathway in the brain and other organs requires proteins such as cathepsin B, irisin, insulin-like growth factor 1 (IGF-1), peroxisome proliferator-activated receptor gamma coactivator 1-alpha (PGC1α), beta-hydroxybutyrate (BHB), and vascular endothelial growth factor (VEGF)”. It is not related to the main study aim of the article and can be deleted.

 Text was removed from the main manuscript: ‘The activation and proper functioning of the BDNF signalling pathway in the brain and other organs requires proteins such as cathepsin B, irisin, insulin-like growth factor 1 (IGF-1), peroxisome proliferator-activated receptor gamma coactivator 1-alpha (PGC1α), beta-hydroxybutyrate (BHB), and vascular endothelial growth factor (VEGF).’

  1. The introduction needs to be restructured. For example, the function of BDNF is described in the sixth paragraph, but it is mentioned in the preceding paragraphs.

 The introduction was restructured.

  1. In the third paragraph of introduction, the authors state that “This protein is only 27 kDa, so BDNF in the blood is able to cross the blood-brain barrier freely”. In human neurons, the BDNF transcript is translated into pre-pro-BDNF in the neuronal cell body, which is cleaved into the precursor pro-BDNF. Pro-BDNF is both converted intracellularly and secreted extracellularly. The authors should indicate whether serum BDNF includes precursor BDNF.

BDNF is synthesized as a precurson molecule called pro-BDNF isoform then conversed into mature BDNF by post-translational cleavage, often during secretion. 

Rafieva, L.M.; Gasanov, E.V. Neurotrophin propeptides: biological functions and molecular mechanisms. Curr. Protein. Pept. Sci. 2016, 17:298-305. doi: : 10.2174/1389203716666150623104145

  1. The authors should elaborate more on the relationship between BDNF in serum and BDNF in brain.

Some authors have suggested [ 21Karege et al. 2002, Rasmusen et al. 2009 and Klein et al.2011] that the BDNF level in the blood may reflect its level in the brain. However, the importance of the brain as a source of BDNF might be overestimated relative to peripheral sources in the blood circulation draining the brain. Above mentioned authors did not measure the contribution of platelets or other peripheral sources of BDNF so these finding require complementary investigation. 

Klein, A. B.; Williamson, R.; Santini, M.A.; Clemmensen, C.; Ettrup, A.; Rios, M.; et al. Blood BDNF concentrations reflect brain-tissue BDNF concentrations across species. Int. J. Neuropsychopharmacol. 2011; 14: 347–353.

Rasmussen P, Brassard P, Adser H, Pedersen MV, Leick L, Hart E, Secher NH, Pedersen BK, Pilegaard H. Evidence for a release of brain-derived neurotrophic factor from the brain during exercise. Exp Physiol. 2009 Oct;94(10):1062-9. doi: 10.1113/expphysiol.2009.048512.

  1. The authors state that “There is a positive correlation between the BDNF concentration in the brain and the serum. A BDNF study in a rat model revealed that the correlation between its concentrations in the cortex and serum was > 0.8.” Please provide references.

References: Karege F, Schwald M, Cisse M. Postnatal developmental profile of brain-derived neurotrophic factor in rat brain and platelets. Neurosci Lett. 2002 Aug 16;328(3):261-4. doi: 10.1016/s0304-3940(02)00529-3.

The following sentences were removed:  “There is a positive correlation between the BDNF concentration in the brain and the serum. A BDNF study in a rat model revealed that the correlation between its concentrations in the cortex and serum was > 0.8.” due to the fact that the study examined a relatively small sample (five Male Wistar rats per group). The importance of the brain as a source of BDNF might be overestimated relative to peripheral sources in the blood circulation draining the brain. Karege et al. did not measure the contribution of platelets or other peripheral sources of BDNF so these finding require complementary investigation. 

  1. The authors state that “Recently, a number of clinical studies have reported the advantageous impact of various types of endurance activity (EA) on enhancing BDNF levels in peripheral blood in post-stroke patients.” Please provide references.

References was added into the main manuscript

 Hsu, C.C.; Fu, T.C.; Huang, S.C.; Chen, C.P.; Wang, J.S. Increased serum brain-derived neurotrophic factor with high-intensity interval training in stroke patients: a randomized controlled trial. Ann. Phys. Rehabil. Med. 2020, 64(4), 101385. doi: 10.1016/j.rehab.2020.03.010.

Boyne, P.; Meyrose, C.; Westover, J.; Whitesel, D.; Hatter, K.; Reisman, D.S.; et al. Exercise intensity affects acute neurotrophic and neurophysiological responses poststroke. J. Appl. Physiol. 2019, 126(2), 431-443.doi: 10.1152/japplphysiol.00594.2018.

Ploughman, M.; Eskes, G.A.; Kelly, L.P.; Kirkland, M.C.; Devasahayam, A.J.; Wallack, E.M.; et al. Synergistic benefits of combined aerobic and cognitive training on fluid intelligence and the role of IGF-1 in chronic stroke. Neurorehabil. Neural. Repair. 2019, 33(3), 199-212.  doi: 10.1177/1545968319832605.

Figure1: The picture is not clear enough.

The figure 1 was changed.

Inconsistent annotation font: TABLE2: “TC - total citation by Web of Science Core Collection, AC - average citations per year by Web of Science Core. Collection, n/a - not available.

The annotation was changed.

Reviewer 2 Report

Even though some technical aspects of the manuscript should be improved, it will be worthy of publication in Journal of Clinical Medicine after major revisions. The manuscript presents an interesting approach towards evaluating the role of endurance training on serum BDNF levels in post-stroke patients by reporting on seven intervention studies. The biggest value of this MS lies in the fact that the authors evaluated the methodological quality of the included studies by using the Physiotherapy Evidence Database (PEDro) criteria. This is a welcomed addition to the systematic reviews within field and, as such, will be of interest to a great majority of readers. On top of reporting on quantitative data obtained through these studies, it is valuable to note that the authors also included information on the quality of each individual study, when assessed within the context of objective criteria.

The manuscript starts by providing an appropriate amount of background data on the financial, social and health burden of stroke. The Materials and Methods section contains well-defined inclusion and exclusion criteria as they pertain to studies included in this systematic review. The authors also chose to represent the general data on the journal placement and average citations which is welcomed. The Results section contains some important data which might be of great interest to a wide audience, but the means of presenting those data are inappropriate and take away from the value of the MS. The Discussion section is well structured but lacks references pertaining some of the statements made.

Therefore, even though the motivation behind this manuscript is clear and will be of value within the field, the manuscript contains many instances of dubious wording, inaccurate sentence constructs, inappropriate reference placement and assorted formatting issues. Of note, these include references to review papers instead of original articles, illegible data tables and major syntax errors in many of the sections throughout the MS. All the comments and suggestions for major revision of this manuscript, as they pertain to each of the sections, can be found within the attached PDF.

Equally important, the manuscript should undergo major editing of the English language and style since many mistakes significantly impact its legibility.

Author Response

Thank you for giving us the opportunity to submit a revised draft of the manuscript ‘The Effect of Endurance Training on Serum BDNF Levels in the Chronic Post-Stroke Phase: Current Evidence and Qualitative Systematic Review’ for publication in the Journal of Clinical Medicine. We appreciate the time and effort that you dedicated to providing feedback on our manuscript and are grateful for the insightful comments on and valuable improvements to our paper. We have incorporated most of the suggestions made by you. Those changes are highlighted in yellow within the manuscript. 

Even though some technical aspects of the manuscript should be improved, it will be worthy of publication in the Journal of Clinical Medicine after major revisions. The manuscript presents an interesting approach towards evaluating the role of endurance training on serum BDNF levels in post-stroke patients by reporting on seven intervention studies. The biggest value of this MS lies in the fact that the authors evaluated the methodological quality of the included studies by using the Physiotherapy Evidence Database (PEDro) criteria. This is a welcomed addition to the systematic reviews within the field and, as such, will be of interest to a great majority of readers. On top of reporting on quantitative data obtained through these studies, it is valuable to note that the authors also included information on the quality of each individual study, when assessed within the context of objective criteria.

Thank you for this comment.

The manuscript starts by providing an appropriate amount of background data on the financial, social and health burden of stroke. The Materials and Methods section contains well-defined inclusion and exclusion criteria as they pertain to studies included in this systematic review. The authors also chose to represent the general data on the journal placement and average citations which is welcomed. The Results section contains some important data which might be of great interest to a wide audience, but the means of presenting those data are inappropriate and take away from the value of the MS. The Discussion section is well structured but lacks references pertaining to some of the statements made.

Thank you for this comment. Text was added into the main manuscript.

Therefore, even though the motivation behind this manuscript is clear and will be of value within the field, the manuscript contains many instances of dubious wording, inaccurate sentence constructs, inappropriate reference placement and assorted formatting issues. Of note, these include references to review papers instead of original articles, illegible data tables and major syntax errors in many of the sections throughout the MS. All the comments and suggestions for major revision of this manuscript, as they pertain to each of the sections, can be found within the attached PDF.

Thank you for suggesting changes. Text was added into the main manuscript.

Equally important, the manuscript should undergo major editing of the English language and style since many mistakes significantly impact its legibility.

Manuscript text has undergone English language editing. 

Round 2

Reviewer 1 Report

The authors have addressed my comments, and I recommend publication of this papre in your journal in its present form.

Reviewer 2 Report

The authors have greatly improved the manuscript by taking into account all of the given suggestion and comments. The addition of multiple new references throughout the MS, reformatting of included tables and replacement of the figure with the one of a greater quality are all now adding to the value of the manuscript, making it clearer and unambiguous.

Four small remarks can be found in the attached PDF that should be corrected before the publication. These include rephrasing two statements, adding a few more references and checking the formatting of Tables 3 and 4.
